# Numerical Solution of the Electrokinetic Equations for Multi-ionic Electrolytes Including Different Ionic Size Related Effects

**DOI:** 10.3390/mi9120647

**Published:** 2018-12-07

**Authors:** José J. López-García, José Horno, Constantino Grosse

**Affiliations:** 1Departamento de Física, Universidad de Jaén, Campus Las Lagunillas, Ed. A-3, 23071 Jaén, Spain; jjgarcia@ujaen.es; 2Departamento de Física, Universidad Nacional de Tucumán, Av. Independencia 1800, 4000 San Miguel de Tucumán, Argentina; cgrosse@herrera.unt.edu.ar

**Keywords:** standard electrokinetic model, ionic size differences effect, conductivity increment, electrophoretic mobility, dielectric increment

## Abstract

One of the main assumptions of the standard electrokinetic model is that ions behave as point-like entities. In a previous work (López-García, et al., 2015) we removed this assumption and analyzed the influence of finite ionic size on the dielectric and electrokinetic properties of colloidal suspensions using both the Bikerman and the Carnahan–Starling equations for the steric interactions. It was shown that these interactions improved upon the standard model predictions so that the surface potential, electrophoretic mobility, and the conductivity and permittivity increment values were increased. In the present study, we extend our preceding works to systems made of three or more ionic species with different ionic sizes. Under these conditions, the Bikerman and Carnahan–Starling expressions cease to be valid since they were deduced for single-size spheres. Fortunately, the Carnahan–Starling expression has been extended to mixtures of spheres of unequal size, namely the “Boublik–Mansoori–Carnahan–Starling–Leland” (BMCSL) equation of state, making it possible to analyze the most general case. It is shown that the BMCSL expression leads to results that differ qualitatively and quantitatively from the standard electrokinetic model.

## 1. Introduction

The standard electrokinetic model is the theory most widely used to characterize electrokinetic phenomena. Although highly versatile and relatively simple to compute, the classical model fails to explain many experimental observations [1,2,3,4]. Because of this, various attempts have been made to modify this model so that the finite ionic size can be taken into account (see Reference [5] and references therein). These corrections to the classic theory strongly improved the agreement between the theoretical predictions and the experimental evidence [6,7,8,9].

Inclusion of the ionic size-related effects is a formidable challenge from the computational point of view. Because of this, most works use the Bikerman [10] or Carnahan–Starling [11] expressions to calculate the steric interactions among ions and only consider binary electrolyte solutions. While these expressions were deduced for single-size spheres, they can be used in this simplest case since, for moderate and high electric potential values, the diffuse electric double layers are populated almost exclusively by counterions so that a single ionic size determines their properties [12,13,14]. However, real systems are often composed by several ionic species. Fortunately, the Carnahan–Starling expression has been extended to mixtures of spheres of unequal size, namely the “Boublik–Mansoori–Carnahan–Starling–Leland” (BMCSL) equation of state [15,16], making it possible to analyze the most general case.

In a previous work [13], we compared the corrections introduced into the standard electrokinetic model by the Bikerman and the Carnahan–Starling expressions. In the present study, we extend our preceding work to systems made of three or more ionic species with different ionic sizes. We show that the use of the BMCSL equation of state leads to results that differ quantitatively and even qualitatively from the standard electrokinetic model.

It is difficult to compare theoretical predictions with experimental results because colloidal suspensions include at least one unknown parameter: the surface charge density or the surface potential. This parameter is determined by fitting a theoretical model to the measured magnitude, which is usually the electrophoretic mobility or the low-frequency dielectric dispersion amplitude. However, when the standard model is used for this purpose, the surface potential values obtained from these two measurements do not coincide. Actually, in some cases, the measured electrophoretic mobility even surpasses the theoretical maximum making any fitting impossible. This leads to the conclusion that both the electrophoretic mobility and the low-frequency dielectric dispersion amplitude values predicted by the standard model are too low [17,18,19]. As already shown in Reference [13] and confirmed in the present work, finite ionic size effects always increase these magnitudes improving the agreement between theoretical and experimental results.

From the practical standpoint, the surface potential (which in the framework of the here considered models coincides with the Zeta potential) is a crucial parameter in all sorts of industrial and scientific applications involving colloids. The determination of this magnitude mainly relies on electrophoretic mobility measurements that are converted into Zeta potential values by means of a theoretical model. Currently, the standard electrokinetic model is used either in its approximate forms (Smoluchowski or Henry equations), rigorous solution (O’Brien and White), or its extension including Stern layer conductivity (Mangelsdorf and White) [20]. Therefore, all ionic size effects are ignored so that ions are only characterized by their valence and diffusion coefficient. We show how the Zeta potential values so determined can be corrected taking into account the finite ionic sizes.

## 2. Theoretical Model

Let us to consider a spherical particle of radius a and surface charge $\sigma_{s}$ immersed in an infinite electrolyte solution with *m* ionic species. The equations governing the dynamics of this system are well known [5,13]:

(a) Modified Nernst–Planck equations for the ionic fluxes:(1)$$c_{i}{\vec{v}}_{i}=-D_{i}c_{i}\nabla \left\{\ln\left[\gamma_{i}c_{i}\right]+\frac{z_{i}e}{kT}\phi \right\}+c_{i}\vec{v}$$

(b) Continuity equations for each ionic species: (2)$$\nabla \cdot \left[c_{i}{\vec{v}}_{i}\right]=-\frac{\partial c_{i}}{\partial t}$$

(c) Poisson equation: (3)$$\nabla^{2}\phi =-\frac{eN_{A}\sum_{i=1}^{m}z_{i}c_{i}}{\varepsilon_{ex}}$$

(d) Navier–Stokes equation for a viscous fluid: (4)$$-\eta \nabla^{2}\vec{v}+\nabla P+eN_{A}\left[\sum_{i=1}^{m}z_{i}c_{i}\right]\nabla \phi +\rho_{f}\left\{\frac{\partial \vec{v}}{\partial t}+\left[\vec{v}\cdot \nabla \right]\vec{v}\right\}=0$$

(e) Continuity equation for an incompressible fluid: (5)$$\nabla \cdot \vec{v}=0$$
where ${\vec{v}}_{i}$, $\gamma_{i}$, $c_{i}$, $z_{i}$, and $D_{i}$ are, respectively, the velocity, the activity coefficient, the local concentration (in mol per unit volume), the signed valence, and the diffusion coefficient of the ionic species *i*. The electric potential is represented by means of the symbol ϕ, $\vec{v}$ is the fluid velocity, and P is the pressure. The constant e represents the elementary charge, while *k*, *T*, *N_A_*, η, $\rho_{f}$, and $\varepsilon_{ex}$ are, respectively, the Boltzmann constant, the absolute temperature, the Avogadro number, the fluid viscosity coefficient, the density of the fluid, and the absolute permittivity of the solution.

For a hypothetic ideal electrolyte solution ($\gamma_{i}$ = 1), this set of equations, together with the appropriate boundary conditions, constitutes the standard electrokinetic model [1,2,3,4]. In order to treat non-ideal solutions, the mathematical form for the activity coefficient of each ionic species must be still specified. We use a local density model, or more precisely, a model that depends on the local concentrations of the different species of solvated ions, namely the BMCSL expression for this coefficient, which accurately describes the thermodynamic properties of hard sphere mixtures [15,16]: (6)$$\begin{array}{cc} \ln\gamma_{i} & =-\left(1-12R_{i}^{2}\frac{\xi_{2}^{2}}{\xi_{3}^{2}}+16R_{i}^{3}\frac{\xi_{2}^{3}}{\xi_{3}^{3}}\right)\ln\left(1-\xi_{3}\right)+\frac{2R_{i}\left(3\xi_{2}+6R_{i}\xi_{1}+4R_{i}^{2}\xi_{0}\right)}{1-\xi_{3}}\\ & +\frac{12R_{i}^{2}\xi_{2}\left(\xi_{2}+2R_{i}\xi_{1}\xi_{3}\right)}{\xi_{3}{\left(1-\xi_{3}\right)}^{2}}-\frac{8R_{i}^{3}\xi_{2}^{3}\left(\xi_{3}^{2}-5\xi_{3}+2\right)}{\xi_{3}^{2}{\left(1-\xi_{3}\right)}^{3}} \end{array}$$
where
(7)$$\xi_{j}=\frac{2^{j-1}\pi N_{A}}{3}\sum_{i=1}^{m}c_{i}R_{i}^{j}\;j\in \left\{0,1,2,3\right\}$$
and $R_{i}$ is the effective hydrated radius of the ionic species *i* (with *i* = 1,…,*m*).

In contrast to the standard electrokinetic model, the modified model contains an extra term (the BMCSL expression for the activity coefficient), representing the impact of the finite size of ions on ionic transport. Note that since different ions have different radii, their Stokes mobilities *ϑ_i_* must be different, and therefore, different ionic species must have unequal diffusion coefficients (*D_i_* = *kTϑ_i_*).

As usual, the equation system is first solved in equilibrium and then under the action of an externally applied AC electric field. Using the assumption that its strength is sufficiently small, the equations are linearized with respect to the field amplitude. The resulting equation system together with the appropriate boundary conditions constitutes the theoretical model that includes the ionic size-related effects (for further details we refer the reader to Reference [13]). This modified model can be numerically solved yielding the electric potential, ion concentrations, and ion and fluid velocity distributions for a rigid spherical particle immersed in a general electrolyte solution. The numerical calculations were performed using the network simulation method [21]. 

## 3. Results and Discussion

In what follows we present numerical results obtained for the theoretical model presented in the previous section, which we shall designate as Standard + BMCSL. The calculations were performed considering a spherical colloidal particle in contact with different aqueous electrolyte solutions specified in each case. The remaining system parameters are given in Table 1, where $\varepsilon_{in}$ is the dielectric permittivity of the particle.

### 3.1. Equilibrium Relationships 

Although the equilibrium properties of the system have been extensively studied in previous papers [22,23], we briefly describe the more important aspects needed for the interpretation of the out of equilibrium system behavior. 

Figure 1 represents the counterion profiles close to the solid–liquid interface calculated for three different electrolyte solution compositions. Two of them are binary and one mixed, chosen in such a way that in all the considered cases the total bulk ionic concentration is kept constant. The standard model exhibits the characteristic excessive growth of the counterion concentrations close to the interface. The profiles for the two binary electrolytes are superimposed since the corresponding Li^+^ and Cs^+^ bulk concentrations coincide, and the ionic valences are also the same (the only difference is in the diffusion coefficient values, but these have no incidence in the equilibrium behavior). On the contrary, for the mixed electrolyte, the Li^+^ profile is always higher than the Cs^+^ one, the corresponding concentrations differing at all points by a constant factor: the ratio of the bulk concentrations 60/20. The Standard + BMCSL behavior is more complex due to the presence of the additional steric force acting on the counterions. In all cases this force limits the excessive concentration buildup close to the interface: none of the considered profiles ever attains unreasonable values. Moreover, for the two binary electrolytes the Li^+^ and Cs^+^ profiles no longer coincide due to the ionic size dependence of the steric force: it is stronger for the larger Li^+^ ion leading to a thicker diffuse double layer than for the smaller Cs^+^. As for the mixed electrolyte, the Li^+^ concentration is higher than the Cs^+^ one only far from the interface where the steric forces become negligible. On the contrary, close to the particle the stronger repelling force acting on the Li^+^ ion leads to a slower growth so that its concentration is surpassed by the smaller Cs^+^ ion that tends to expel the Li^+^ ions from the immediate vicinity of the interface. 

Note that this and all the following figures were calculated using the surface charge on the suspended particle rather than its surface potential as a system parameter. This is a necessary choice due to the dependence of the relationship between these two magnitudes on the theoretical model used for its calculation. This is shown in Figure 2 for the same electrolyte solutions and theoretical models as in Figure 1. It shows the tremendous increase of the surface potential value with the steric forces at any given surface charge value, which is due to the increase of the electric double layer thickness. Thus, Figure 2 shows that a comparison of different models using a surface potential of −150 mV, for example, would involve a particle with a surface charge of −0.4 C/m^2^ for the standard and of −0.125 C/m^2^ for the Standard + BMCSL models. Therefore, a comparison of the predictions of these two models at any given surface potential value would involve totally different suspended particles: with a much higher charge in the standard model case. Moreover, Figure 2 shows that even a comparison involving just the Standard + BMCSL model predictions with different electrolyte solutions requires the use of the surface charge as system parameter rather than the surface potential.

### 3.2. AC Behavior

We now consider the system response under the action of an applied AC electric field with amplitude $E_{a}$ and frequency f. 

#### 3.2.1. Dielectric Response

The dielectric response of the system is determined computing the frequency behavior of the dipolar coefficient: the amplitude (divided by $E_{a}$) of the field induced dipolar field far away from the particle:(8)$$d^{*}(f)=d^{\prime }(f)+id^{\prime \prime }(f)$$
where the asterisk denotes a complex magnitude. The spectra of the real and imaginary parts of this coefficient are represented in Figure 3 and Figure 4.

For uncharged particles, the low-frequency dipolar coefficient value is simply equal to −1/2 (this is rigorous for point ions but also holds to high approximation for finite size ions because the ionic concentrations remain everywhere low when particles are uncharged). For increasing (in absolute value) surface charge values, the standard model behavior can be analyzed using the results presented in Reference [24]. When the counterion and co-ion diffusion coefficients differ from one another, volume charge density clouds buildup outside the equilibrium double layer boundaries. This charge that has the same sign as $D_{counterion}-D_{coion}$ increases the dipolar coefficient value which, for equal diffusion coefficients, is solely due to the charge density distribution inside the double layer. This explains why $Re({d^{*}}_{\mathrm{CsCl}})>Re({d^{*}}_{\mathrm{LiCl}})$ in Figure 3. As for the dipolar coefficient corresponding to the mixed electrolyte, it always lies in between the two binary cases closer to LiCl than to CsCl as expected.

The Standard + BMCSL theory predicts a very strong increase of the low frequency dipolar coefficient value as compared to the standard model. The main reason for this is the increment of the diffuse double layer thickness, Figure 1, that increases the distance of the charged fluid away from the zero-velocity boundary condition at the particle surface. This leads to an enhancement of the convective flow contribution to its dipolar coefficient [25]. Besides this difference, the Standard + BMCSL low-frequency dipolar coefficient behavior is similar to that of the standard model.

At higher frequencies (10^6^–10^7^ Hz), above the low-frequency dielectric dispersion region, the concentration polarization vanishes, and the charged particle behaves essentially as a conductive sphere inside a conductive medium with different conductivity value. Its dipolar coefficient tends to the classical value:(9)$$d=\frac{K_{in}-K^{\infty }}{K_{in}+2K^{\infty }}$$
where
(10)$$K_{in}=\frac{2\lambda }{a}$$
is the equivalent conductivity of the insulating particle of radius a surrounded by a conducting layer with surface conductivity λ [26]. For highly-charged particles, the dipolar coefficient tends to a maximum value *d* → 1, that is far from being attained in the considered case, Figure 3. The standard model curves in this figure clearly show the increment of the surface conductivity with the counterion diffusion coefficient. As for the BMCSL model, it predicts a much stronger increment that further depends on the finite ionic sizes that increase the diffuse double layer thickness. Figure 3 clearly shows that the highest surface conductivity value corresponds to the binary LiCl electrolyte in which the lowest diffusion coefficient of the Li^+^ counterion is compensated by the largest diffuse double layer thickness, Figure 1.

At even higher frequencies (10^9^–10^10^ Hz), the field induced surface charge densities can no longer build up so that only polarization charges contribute to the dipolar coefficient. The charged particle behaves, therefore, as a low permittivity insulating sphere in a high permittivity medium. Its classical dipole coefficient value is:(11)$$d=\frac{\varepsilon_{in}-\varepsilon_{ex}}{\varepsilon_{in}+2\varepsilon_{ex}}=\frac{2-80}{2+160}\approx -\frac{1}{2}$$

As expected, the imaginary part of the dipolar coefficient tends to zero at low frequencies independently of the considered model. Moreover, another feature that can only be appreciated in a Log-Log plot is that $d^{\prime \prime }$ becomes proportional to the frequency when f→0. This property will be used in the interpretation of the suspension permittivity. At higher frequencies, each $d^{\prime \prime }$ curve attains a maximum with a value that is proportional to the relaxation amplitude of the real part of the dipolar coefficient, as expected. At high frequencies, the standard model predicts the highest relaxation frequency for CsCl and the lowest for LiCl as expected in view of the higher diffusion coefficient of Cs^+^ as compared to Li^+^. On the contrary, according to the Standard + BMCSL model, this qualitative behavior is reversed showing again that the surface conductivity of the particle immersed in binary LiCl is higher than in CsCl because the effect of the greater double layer thickness in the former case outweighs that of the higher diffusion coefficient in the latter. Finally, at even higher frequencies, the imaginary part of the dipolar coefficient tends to zero for both considered models, as expected. 

The complex conductivity of the suspension can now be deduced from the dipolar coefficient spectra:(12)$$K^{*}(f)=K(f)+i2\pi f\varepsilon (f)=K_{ex}^{*}(f)\left[1+3\varphi d^{*}(f)\right]$$
where
(13)$$K_{ex}^{*}(f)=K^{\infty }+i2\pi f\varepsilon_{ex}$$
is the complex conductivity of the electrolyte solution and φ is the volume fraction occupied by the particles in the suspension (assumed to be low for Equation (12) to be valid). Combining Equations (8), (12), and (13) leads to the suspension permittivity and conductivity expressions:(14)$$\varepsilon (f)=\varepsilon_{ex}\left\{1+3\varphi \left[d^{\prime }(f)+\frac{K^{\infty }}{\omega \varepsilon_{ex}}d^{\prime \prime }(f)\right]\right\}$$
(15)$$K(f)=K^{\infty }\left\{1+3\varphi \left[d^{\prime }(f)-\frac{\omega \varepsilon_{ex}}{K^{\infty }}d^{\prime \prime }(f)\right]\right\}$$

These equations make it possible to define the permittivity and conductivity increments, which are independent of the particle volume fraction:(16)$$\Delta \varepsilon (f)\equiv \frac{\varepsilon (f)-\varepsilon_{ex}}{\varphi \varepsilon_{0}}=3\frac{\varepsilon_{ex}}{\varepsilon_{0}}\left[d^{\prime }(f)+\frac{K^{\infty }}{2\pi f\varepsilon_{ex}}d^{\prime \prime }(f)\right]$$
(17)$$\Delta K(f)\equiv \frac{K(f)-K^{\infty }}{\varphi K^{\infty }}=3\left[d^{\prime }(f)-\frac{2\pi f\varepsilon_{ex}}{K^{\infty }}d^{\prime \prime }(f)\right]$$
The corresponding spectra are represented in Figure 5a,b and Figure 6.

Figure 5a shows the permittivity increment spectra for low frequencies. Obviously, the huge values of the permittivity increment imply that they are mainly due to the second addend in Equation (16): quotient of the imaginary part of the dipolar coefficient and frequency (the imaginary part of the dipolar coefficient is proportional to the frequency for f→0). As can be seen, according to the Standard model the dielectric increment is highest (lowest) when the counterion diffusion coefficient is lower (higher) than that of the co-ion: LiCl (CsCl). The main reason for this behavior is the increase of the second addend in Equation (16) with the characteristic time of the low-frequency dielectric dispersion, which is determined by the ion with the lowest diffusion coefficient: Li^+^ [27]. As for the Standard + BMCSL predictions, they show the same qualitative behavior as for the standard model except for a much higher amplitude. 

This difference is mainly due to the value of the characteristic time of the low-frequency dielectric dispersion that increases with the convective fluid flow around the particle (Equation (38) in Reference [25]). This is also the reason why the permittivity increment of the mixed electrolyte tends to that of the binary CsCl electrolyte for large surface charges: under these conditions the smaller Cs^+^ ions expel the larger Li^+^ from the diffuse double layer, Figure 1.

Figure 5b shows the permittivity increment spectra for high frequencies, for which the dispersion amplitude is so small as compared to the low-frequency dielectric dispersion that they would be invisible if Figure 5a were extended up to 10^10^ Hz. This Maxwell–Wagner dispersion confirms the comments following Figure 3. For the standard model, Equations (9) and (10) show that the highest (lowest) dispersion amplitude corresponds to the CsCl (LiCl) binary electrolyte in view of the high (low) diffusion coefficient of the Cs^+^ (Li^+^) ion. For the Standard + BMCSL model, the dispersion amplitude is much greater because steric forces among ions increase the thickness of the diffuse double layer leading to a large increase of the surface conductivity. This effect is so strong that it outweighs that determined by the diffusion coefficient value: the dispersion amplitude is highest (lowest) for LiCl (CsCl) binary electrolyte. Finally, Figure 5b clearly shows the increase of the relaxation frequency predicted by the Standard + BMCSL as compared to the standard model, which is due to the increment of the surface conductivity.

Figure 6 shows the conductivity increment spectra for the considered systems and models. At low frequencies, the imaginary part of the dipolar coefficient in Equation (17) vanishes so that the conductivity increment is just proportional to the real part of the dipolar coefficient, Figure 3. This dependence remains practically throughout the low-frequency dielectric dispersion range because the second addend in Equation (17) contains the frequency as a factor that is still small. On the contrary, in the high-frequency range corresponding to the Maxwell–Wagner dispersion this factor becomes decisive. The huge increment of the conductivity at high frequencies is almost entirely due to the imaginary part of the dipolar coefficient. Classically, in this frequency range its value should decrease as 1/f so that its product by the frequency leads to a constant high amplitude conductivity increment value:(18)$$\Delta K(f\to \infty )=3\left\{d^{\prime }(f\to \infty )-\frac{2\pi \varepsilon_{ex}}{K^{\infty }}\lim_{f\to \infty }\left[fd^{\prime \prime }(f)\right]\right\}$$

Actually, all the conductivity increment curves suddenly decrease on the right-hand side of Figure 6. This is due to inertial effects: the last addend in Equation (4), which contains the fluid mass density and opposes convective fluid velocity changes, becomes non-negligible at the highest frequencies. 

#### 3.2.2. Electrophoretic Mobility

The dimensionless electrophoretic mobility is determined calculating the field induced fluid velocity $v_{\infty }^{*}(f)$ far away from the suspended particle:(19)$$\mu^{*}(f)=\frac{3e\eta v_{p}^{*}(f)}{2\varepsilon_{e}kTE_{a}}$$
where $v_{p}^{*}=-v_{\infty }^{*}$ is the electrophoretic velocity of the particle. 

Figure 7 represents the spectra corresponding to the real part of the electrophoretic mobility. This magnitude is determined by the total fluid flow in the diffuse double layer which, at low frequencies, is caused by electroosmosis and capillary osmosis. The first is proportional to the total tangential field in the double layer, $E_{a}\left(1-d^{*}\right)$ so that, in the considered case, it slightly decreases with frequency, Figure 3. The second is proportional to the tangential gradient of the electrolyte concentration and opposes the fluid flow [27]. It decreases with frequency over the low-frequency dielectric dispersion range and vanishes at around 10^6^–10^7^ Hz. 

The increase (in modulus) of the electrophoretic mobility at low frequencies shows that in the considered case the electroosmotic contribution outweighs that of the dipolar coefficient. At higher frequencies, 10^8^–10^9^ Hz, the dipolar coefficient strongly diminishes, Figure 3, which should lead to an increase the electrophoretic mobility. This behavior does not appear in Figure 7, since at these high-frequency values the fluid flow behavior is dominated by inertial effects that lead to a monotonic decrease of the electrophoretic mobility. 

As for the different considered electrolytes, Figure 7 shows the expected behavior predicted by the standard model. At low frequencies, the electrophoretic mobility modulus is highest (lowest) for CsCl (LiCl) in view of the higher (lower) diffusion coefficient of the Cs^+^ (Li^+^) ion that leads to a higher (lower) fluid velocity [27]. The behavior predicted by the Standard + BMCSL model is qualitatively similar to that of the standard model but with much greater mobility values (in modulus). Again, this difference originates in the increase of the diffuse double layer thickness with the ionic size, Figure 1, which increases the fluid flow around the particle. 

However, this is not a trivial conclusion as can be seen comparing the results of this work with those obtained considering the standard model with the presence of stagnant layer conductivity [28]. As can be seen, there is a striking similarity between the dipolar coefficient and the permittivity and conductivity increment spectra when the surface conductivity is enhanced by either an increase of the double layer thickness or the presence of an anomalous surface conductivity. On the contrary, the mobility results are completely different: this magnitude increases with the ionic size but decreases with the anomalous surface conductivity. The reason for this contrasting behavior is that the dielectric response depends on the electric current density while the electrophoretic mobility depends on the fluid flow. Both magnitudes increase with the ionic size while the anomalous surface conductivity only increases the current density since fluid flow is not allowed inside the stagnant layer. Therefore, the current density in the stagnant layer decreases the (1 − *d**) coefficient leading to a decrement of the electrophoretic mobility.

Figure 8 represents the electrophoretic mobility value calculated in the limit f→0 as a function of the surface charge. The standard model behavior corresponds to the well-known results calculated in Reference [29], except for the figure inversion due to the negative sign of the particle charge and to the use of its surface charge rather than its surface potential as an independent variable. The rather weak differences among the three plotted curves are due to the diffusion coefficient values: the fluid flow in the diffuse double layer increases with the counterion diffusion coefficient so that the electrophoretic mobility is highest (in absolute value) for CsCl, lowest for LiCl, and intermediate for the mixed electrolyte solution. The Standard + BMCSL model leads to a strong increase (in absolute value) of the electrophoretic mobility. Again, the main reason for this behavior is the greater thickness of the diffuse double layer that increases the distance of the charged fluid away from the zero-velocity boundary condition at the particle surface. However, two competing effects are present in this case: the Cs^+^ (Li^+^) ion has a high (low) diffusion coefficient but a small (large) size that corresponds to a thin (thick) diffuse double layer. For weakly charged particles, the second effect outweighs the first so that the mixed electrolyte closely follows the binary LiCl behavior. On the contrary, for highly-charged particles, the Li^+^ ions are practically expelled from the interface neighborhood, Figure 1, because of which the mixed electrolyte tends to the binary CsCl behavior.

## 4. Conclusions

In this work we present numerical results for AC dielectric and electrokinetic properties of colloidal suspensions in aqueous multi-ionic electrolytes taking into account finite ionic size differences. This is done combining the standard electrokinetic model with steric interactions among ions represented by means of the “Boublik–Mansoori–Carnahan–Starling–Leland” (BMCSL) equation of state. To the best of our knowledge this is the first time that this problem has been solved out of equilibrium and for two different size counterions.

The obtained spectra are compared to the standard model predictions considering negative colloidal particles suspended in three electrolyte solutions having all the same electrolyte concentration: 80 mM LiCl, 80 mM CsCl, and 60 mM LiCl + 20 mM CsCl.

The standard model shows the expected differences among the obtained spectra differentiated only by the counterion diffusion coefficients: lowest for Li^+^ and highest for Cs^+^. Therefore, the mixed electrolyte solution results always lie in between those of the two binary electrolyte cases. 

On the contrary, the Standard + BMCSL model also takes into account the counterion size difference: largest for Li^+^ and smallest for Cs^+^. Unlike the diffusion coefficient differences, the presence of small counterions in a mixed electrolyte influences the large counterion concentration and vice versa. Therefore, there is no simple qualitative relationship between the mixed electrolyte results and those of the two binary electrolyte cases. Depending on the calculated magnitude, the particle charge, or the AC frequency, these results may lie close to those of LiCl, or CsCl, or even outside the range determined by the two binary electrolytes.

A comparison between the standard and the Standard + BMCSL model results show very large differences due, at least in part, to the large surface charge and electrolyte concentration values used in this study. However, these values were chosen in order to easily appreciate the differences between the results corresponding to each of the considered models since, as previously noted in Reference [13], the inclusion of steric interactions produces non-negligible modifications to the standard model results in all the usual cases. 

It is well known that the standard model is unable to simultaneously provide an interpretation of dielectric and electrokinetic properties: the surface potential value required do fit dielectric data is different from that needed to interpret the electrophoretic mobility value. In order to accomplish this objective both the suspension permittivity and the particle electrophoretic mobility should be larger than predicted by the standard model. A possible solution: the assumption of the existence of the Stern or stagnant layer conductivity failed to satisfy this requirement since it increases the permittivity but decreases the mobility. On the contrary, steric interactions among ions increase both of these magnitudes making it possible to simultaneously provide an interpretation of dielectric and electrokinetic data in many cases. However, such an interpretation requires the use of precise steric interaction calculations so that the use of the BMCSL equation becomes unavoidable for multi-ionic systems.

Even when both dielectric and electrokinetic measurements are unavailable, which is often the case, our results make it possible to directly improve experimental Zeta potential values obtained from electrophoretic mobility measurements using the standard model [20,29]. Figure 9 represents the electrophoretic mobility as a function of the dimensionless surface potential (or Zeta potential ζ, since both magnitudes coincide for the considered models) calculated for different electrolyte concentrations. It is identical to Figure 4 in Reference [29] except for the sign of the particle charge and the chosen electrolyte: LiCl rather than KCl. It shows that if the Zeta potential has been obtained for such a system: κ*a* = 100, ζ = −77 mV (*y* = −3) for example, it follows that the electrophoretic mobility was −5 and, therefore, the Zeta potential value corrected by taking into account the finite ionic sizes is −94 mV (*y* = −3.7). As can be seen, the differences between these two potential values become smaller for weaker potentials but can be much greater for stronger potential values. 

## Figures and Tables

**Figure 1 micromachines-09-00647-f001:**
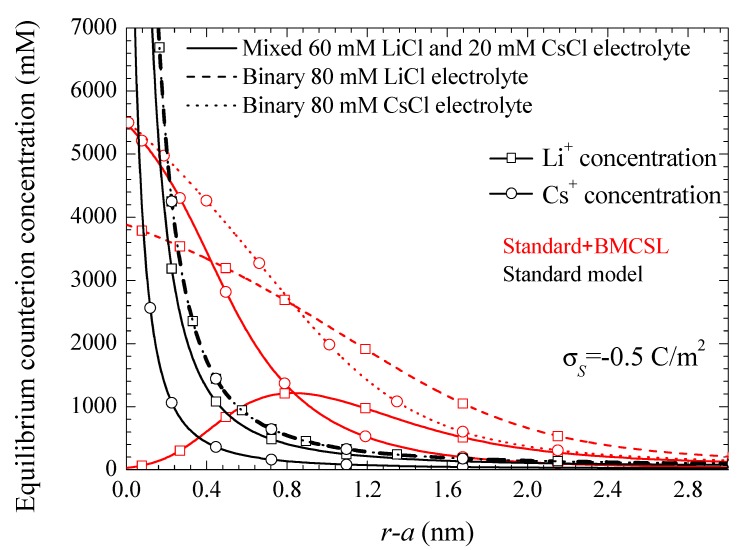
Counterion concentration profiles for the indicated surface charge density and the considered models and electrolyte solutions. Remaining parameters given in Table 1.

**Figure 2 micromachines-09-00647-f002:**
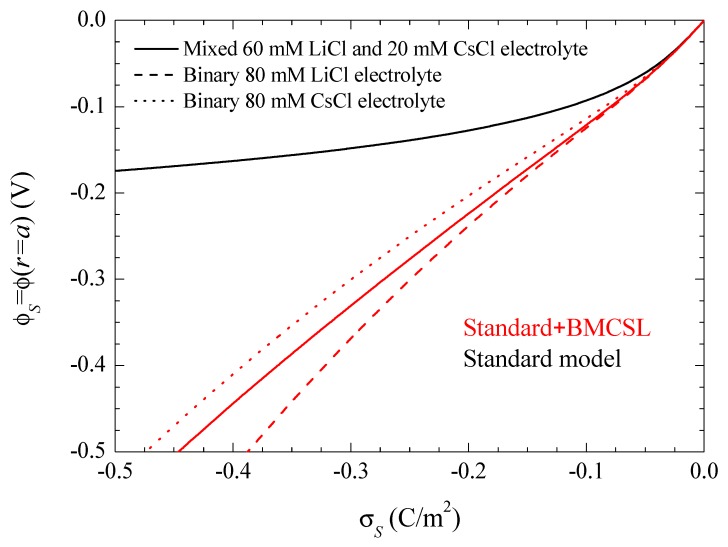
Surface potential dependence on the surface charge density for the considered models and electrolyte solutions. The three standard model curves corresponding to different electrolytes overlap since equilibrium results cannot depend on diffusion coefficient values. Remaining parameters given in Table 1.

**Figure 3 micromachines-09-00647-f003:**
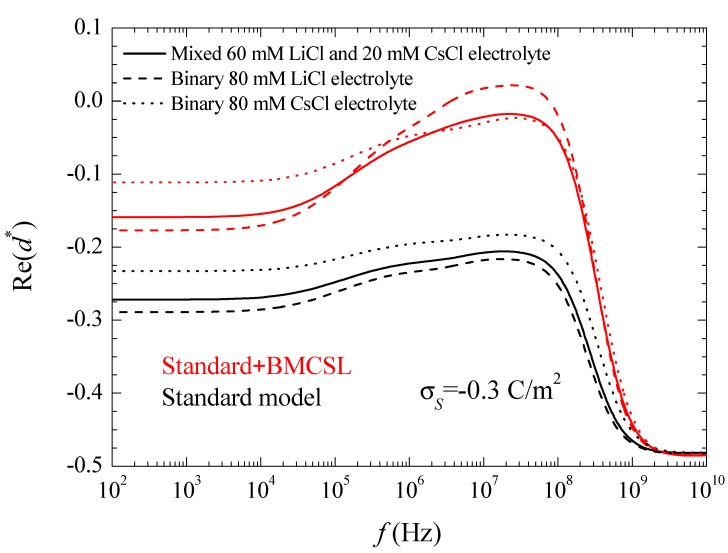
Real part of the dipolar coefficient spectra for the indicated surface charge density and the considered models and electrolyte solutions. Remaining parameters given in Table 1.

**Figure 4 micromachines-09-00647-f004:**
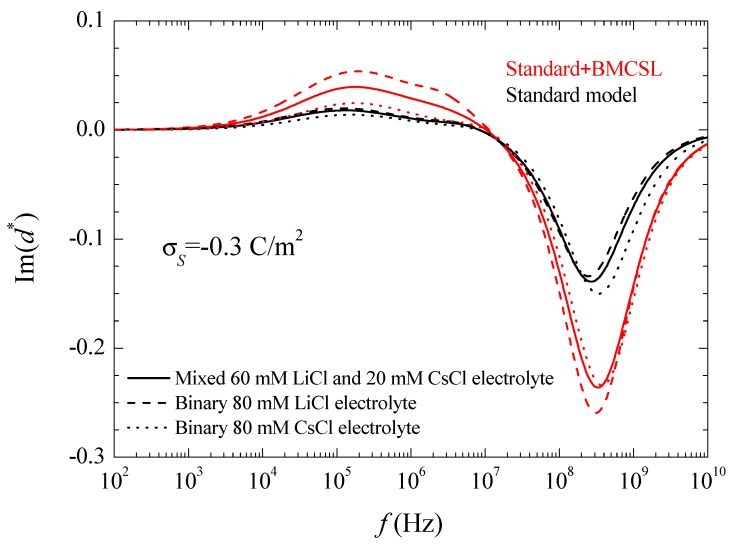
Imaginary part of the dipolar coefficient spectra for the indicated surface charge density and the considered models and electrolyte solutions. Remaining parameters given in Table 1.

**Figure 5 micromachines-09-00647-f005:**
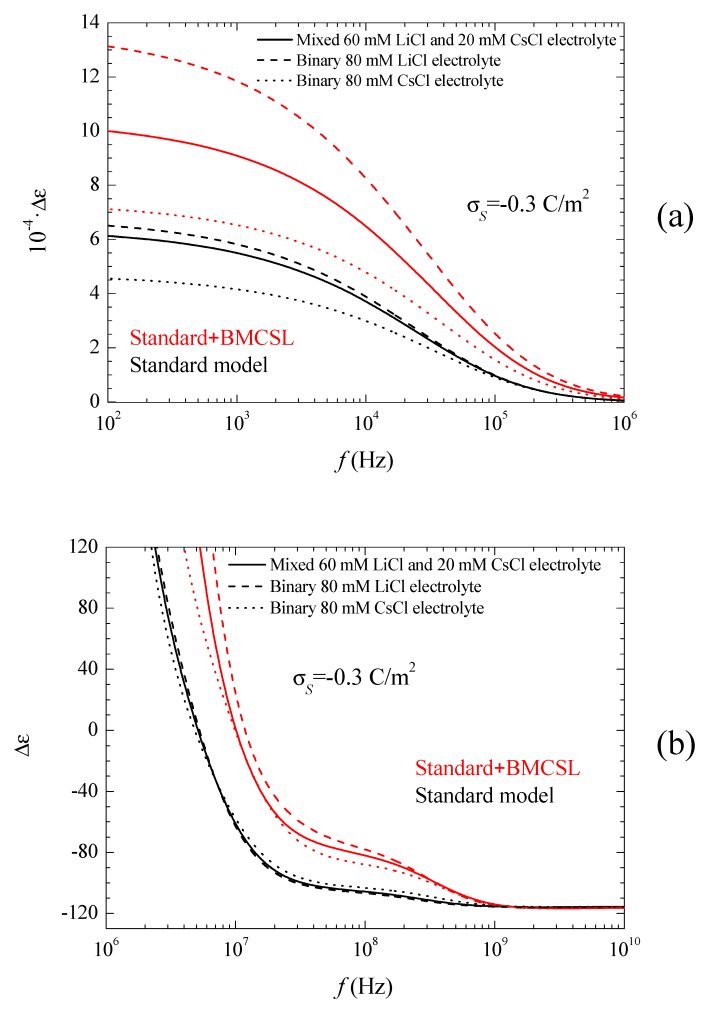
Permittivity increment low (**a**) and high (**b**) frequency spectra for the indicated surface charge density and the considered models and electrolyte solutions. Remaining parameters given in Table 1.

**Figure 6 micromachines-09-00647-f006:**
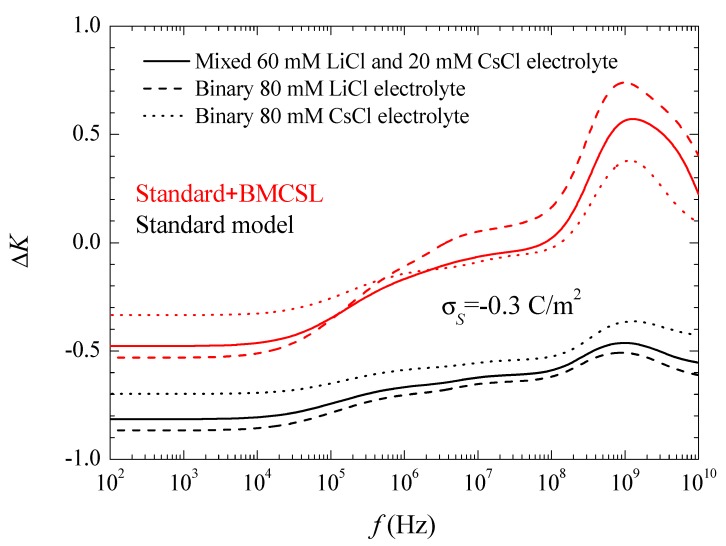
Conductivity increment spectra for the indicated surface charge density and the considered models and electrolyte solutions. Remaining parameters given in Table 1.

**Figure 7 micromachines-09-00647-f007:**
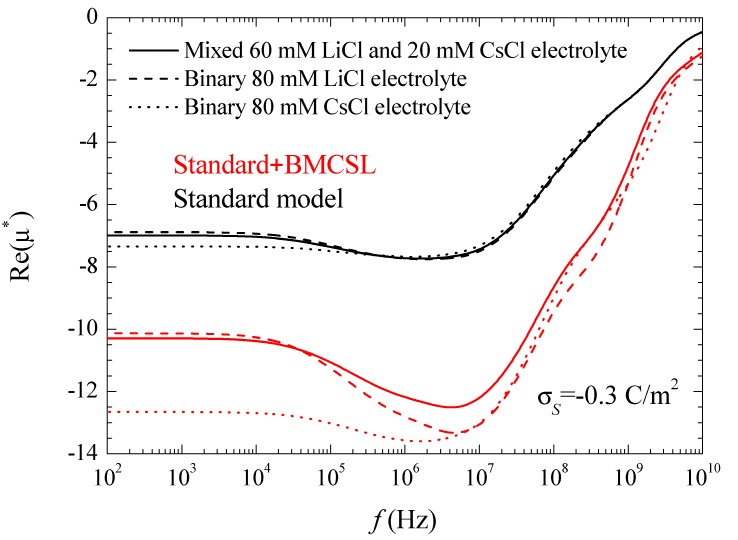
Real part of the dimensionless electrophoretic mobility spectra for the indicated surface charge density and the considered models and electrolyte solutions. Remaining parameters given in Table 1.

**Figure 8 micromachines-09-00647-f008:**
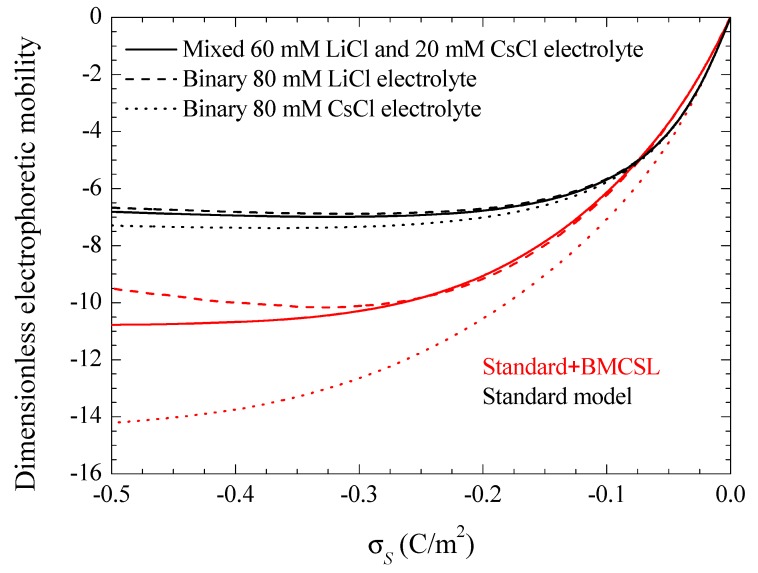
Dimensionless electrophoretic mobility dependence on the surface charge density for the considered models and electrolyte solutions. Remaining parameters given in Table 1.

**Figure 9 micromachines-09-00647-f009:**
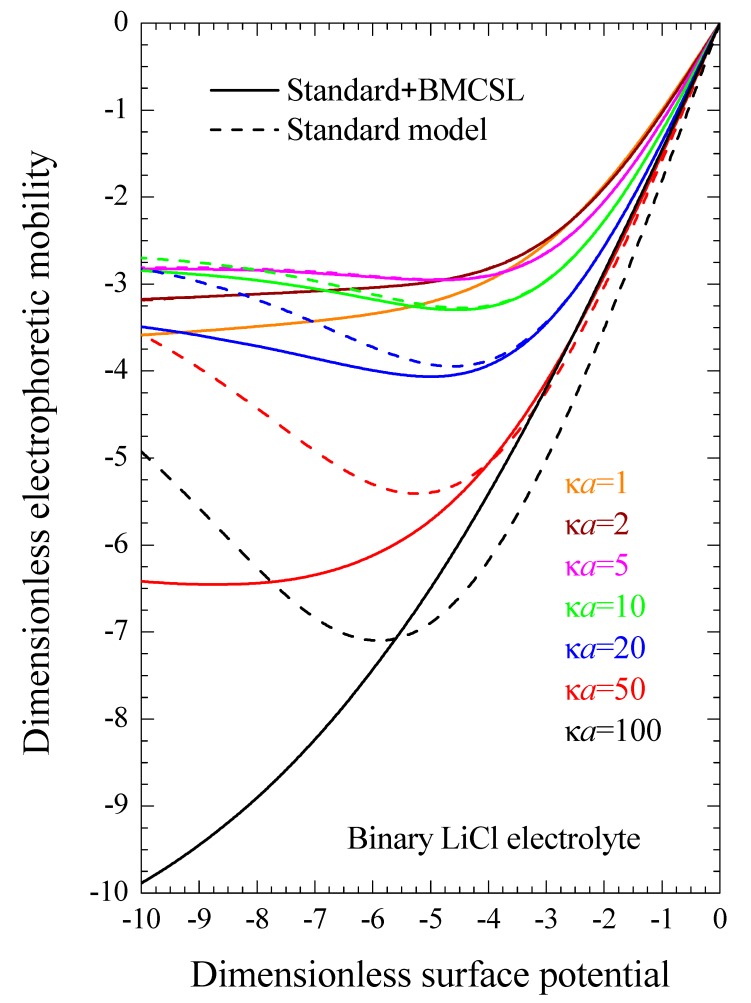
Dimensionless electrophoretic mobility dependence on the dimensionless surface potential $y=\phi_{s}e/kT$ for the considered models and for binary LiCl electrolyte solutions at different concentrations: $\kappa =\sqrt{e^{2}N_{A}\sum_{i=1}^{m}z_{i}c_{i}^{\infty }/kT\varepsilon_{ex}}$. Remaining parameters given in Table 1.

**Table 1 micromachines-09-00647-t001:** Parameter values used in all the simulations.

T=298 K	$\eta =0.89\times 10^{-3}$ P	$\varepsilon_{ex}=80\varepsilon_{0}$
$R_{{\mathrm{Cl}}^{-}}=3.37\;Å$	$D_{{\mathrm{Cl}}^{-}}=2.03\times 10^{-9}\;m^{2}/s$	$\varepsilon_{in}=20\varepsilon_{0}$
$R_{{\mathrm{Cs}}^{+}}=3.29\;Å$	$D_{{\mathrm{Cs}}^{+}}=2.06\times 10^{-9}\;m^{2}/s$	$\frac{1}{2}\sum_{i=1}^{m}z_{i}^{2}c_{i}^{\infty }=80\;\mathrm{mM}$
$R_{{\mathrm{Li}}^{+}}=3.82\;Å$	$D_{{\mathrm{Li}}^{+}}=1.03\times 10^{-9}\;m^{2}/s$	a=100 nm

## Data Availability

The data used to support the findings of this study are available from the corresponding author upon request.

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
