# Peer review of "Numerical Solution of the Electrokinetic Equations for Multi-ionic Electrolytes Including Different Ionic Size Related Effects"

_micromachines, 2018, doi:10.3390/mi9120647_

Round 1

Reviewer 1 Report

Review report on “Numerical solution of the electrokinetic equations for multiionic electrolytes including different ionic size related effects”

The authors present a numerical study using BMCSL expression to analyze systems with different sized ionic species. The article essentially presents an extension of previous work by same authors to study solutions for a more general case. The article is well written and easy to read. The primary finding of the article is the difference observed in the predictions from the presented model and the standard electrokinetic model. I find the study to be scientifically sound and am glad to recommend its publication. Some minor comments for authors to consider:

1.      While the significance of this study from a theoretical standpoint is obvious, perhaps it would be helpful to motivate this study more from an applications standpoint. More specifically, can the authors comment on the real-world scenarios where these more accurate results can be helpful. Notwithstanding, I agree with the significance of this study from a physics perspective.

2.      The authors should consider adding a brief description of how the presented equation system is more suitable when compared against experiments. While the authors have presented a comparison with the standard model, an explicit direct comparison with experimental results is missing.

Author Response

Comments:

1.     While the significance of this study from a theoretical standpoint is obvious, perhaps it would be helpful to motivate this study more from an applications standpoint. More specifically, can the authors comment on the real-world scenarios where these more accurate results can be helpful. Notwithstanding, I agree with the significance of this study from a physics perspective.

ANSWER: This issue is now discussed at the end of the Conclusions and in the new Figure 9.

2.     The authors should consider adding a brief description of how the presented equation system is more suitable when compared against experiments. While the authors have presented a comparison with the standard model, an explicit direct comparison with experimental results is missing.

ANSWER: This issue is now discussed at the end of the Introduction.

Reviewer 2 Report

By combining the Standard Electrokinetic model with steric interactions among ions, numerical results for AC dielectric and electrokinetic properties of colloidal suspensions in aqueous multiionic electrolytes are obtianed and analyzed. The manuscript is well-organized and its results are supported. This manuscript can be accepted after some minor revisions:  

1. Please change all the "Fig. xxx" into "Figure xxx". They need to be consistent. 

2. In Figure 2, are curves of Binary 80 mM LiCl electrolyte and Binary 80 mM CsCl electrolyte not shown or merged together? Please clarify. 

Author Response

Comments:  

1. Please change all the "Fig. xxx" into "Figure xxx". They need to be consistent. 

ANSWER: This has been done.

2. In Figure 2, are curves of Binary 80 mM LiCl electrolyte and Binary 80 mM CsCl electrolyte not shown or merged together? Please clarify. 

ANSWER: The caption of Figure 2 has been modified explaining this overlap.

Reviewer 3 Report

The authors extend their preceding works of the numerical solution of electrokinetic equations to the systems of three or more ionic sizes.  In their preceding work, a combination of Bikerman and the Carnahan-Starling equations was employed for modeling finite ionic size instead of using the standard electrokinetic model. In this paper, an extended Carnahan-Starling, which is also known as Boublik-Mansoori-Carnahan-Starling-Leland (BMCSL) equation, was used for modeling the spheres of unequal sizes.  The key difference is that the BMCSL equation includes an extra term representing the transport of ions with different sizes in Eq.(6). Calculations were performed using different mixings of Cl-, Li+ and Cs+ ions, where the size of Cs+ or Cl- is about the double of Li+. The results show that the use of BMCSL equation leads to the results differ from a standard electrokinetic model. 

Comments to the authors:

1. The theoretical investigation of the electrokinetic equations is very interesting and seems to have great potential. However, the authors are suggested to provide information on the potential applications using the proposed modeling in the second of Introduction, so that the usefulness of the work can be strengthened.

2. Although the results of surface charge, surface potential, permittivity and conductivity show clear differences between the models of standard and standard+BMCSL ones, experimental results are still NEEDED to validate the model, as well as convincing the readers.

3. The authors should still cite proper sources of the equations, as well as carefully defining the variables. For example, this reviewer cannot find the definition of "T" in Eq.(1).

4. The legends should be consistent with the plots in the charts. For example, the boxed/circled/solid/dashed/dotted lines in the legend of Fig.1 are all black while the plottings are in either red or blue. 

Author Response

Comments:

1. The theoretical investigation of the electrokinetic equations is very interesting and seems to have great potential. However, the authors are suggested to provide information on the potential applications using the proposed modeling in the second of Introduction, so that the usefulness of the work can be strengthened.

ANSWER: This issue has been now addressed in the Introduction, new Figure 9, and in the Conclusions.  

2. Although the results of surface charge, surface potential, permittivity and conductivity show clear differences between the models of standard and standard+BMCSL ones, experimental results are still NEEDED to validate the model, as well as convincing the readers.

ANSWER: This issue is now discussed at the end of the Introduction.

3. The authors should still cite proper sources of the equations, as well as carefully defining the variables. For example, this reviewer cannot find the definition of "T" in Eq. (1).

ANSWER: This has been done.

4. The legends should be consistent with the plots in the charts. For example, the boxed/circled/solid/dashed/dotted lines in the legend of Fig.1 are all black while the plottings are in either red or blue. 

ANSWER: Figure 1 has been modified. 

Round 2

Reviewer 3 Report

All the concerns of this reviewer have been well addressed.